# Combined Presence in Heterozygosis of Two Variant Usher Syndrome Genes in Two Siblings Affected by Isolated Profound Age-Related Hearing Loss

**DOI:** 10.3390/biomedicines11102657

**Published:** 2023-09-28

**Authors:** Nica Borgese, Andrés Guillén-Samander, Sara Francesca Colombo, Giulia Mancassola, Federica Di Berardino, Diego Zanetti, Paola Carrera

**Affiliations:** 1Consiglio Nazionale delle Ricerche Neuroscience Institute, 20854 Vedano al Lambro, Italy; sara.colombo@in.cnr.it; 2Bernhard Nocht Institute for Tropical Medicine, 20359 Hamburg, Germany; andres.guillen@bnitm.de; 3NeuroMi Milan Center for Neuroscience, Milan, Italy; 4Unit of Genomics for Human Disease Diagnosis, IRCCS Ospedale San Raffaele, 20132 Milan, Italy; mancassola.giulia@hsr.it; 5Laboratory of Clinical Molecular Genetics, IRCCS Ospedale San Raffaele, 20132 Milan, Italy; 6Audiology Unit, Department of Specialistic Surgical Sciences, Fondazione IRCCS Ca’ Granda, Ospedale Maggiore Policlinico, 20122 Milan, Italy; federica.diberardino@policlinico.mi.it (F.D.B.); diego.zanetti@policlinico.mi.it (D.Z.); 7Department of Clinical Sciences and Community Health, University of Milano, 20122 Milan, Italy

**Keywords:** ADGRV1, haploinsufficiency, nonsyndromic deafness, SANS, Usher gene network

## Abstract

Sensorineural age-related hearing loss affects a large proportion of the elderly population, and has both environmental and genetic causes. Notwithstanding increasing interest in this debilitating condition, the genetic risk factors remain largely unknown. Here, we report the case of two sisters affected by isolated profound sensorineural hearing loss after the age of seventy. Genomic DNA sequencing revealed that the siblings shared two monoallelic variants in two genes linked to Usher Syndrome (*USH* genes), a recessive disorder of the ear and the retina: a rare pathogenic truncating variant in *USH1G* and a previously unreported missense variant in *ADGRV1*. Structure predictions suggest a negative effect on protein stability of the latter variant, allowing its classification as likely pathogenic according to American College of Medical Genetics criteria. Thus, the presence in heterozygosis of two recessive alleles, which each cause syndromic deafness, may underlie digenic inheritance of the age-related non-syndromic hearing loss of the siblings, a hypothesis that is strengthened by the knowledge that the two genes are integrated in the same functional network, which underlies stereocilium development and organization. These results enlarge the spectrum and complexity of the phenotypic consequences of *USH* gene mutations beyond the simple Mendelian inheritance of classical Usher syndrome.

## 1. Introduction

Sensorineural age-related hearing loss (ARHL), a condition caused by inner ear damage, affects a large proportion of the elderly population (World Health Organization https://www.who.int/health-topics/hearing-loss#tab=tab_1 (accessed on 29 March 2023)) [1,2], and depending on the degree of severity, it may lead to social isolation, cognitive deficits and/or depression [3]. Both environmental and genetic risk factors underlie this disability, and the latter’s contribution is estimated at 35–70% in different studies (reviewed in [4]).

Given the recognized genetic basis of ARHL, a number of large-scale population studies have been carried out with the aim of identifying genetic risk factors ([5] and references therein; reviewed in [4,6,7]). The general conclusion of these studies is that ARHL is a complex, highly heterogeneous, and polygenic pathology [8], and that the unequivocally identified genetic risk factors linked to ARHL explain only a portion of the estimated heritability.

Two hypotheses to explain the modest outcome of Genome-Wide Association Studies (GWAS) are (i) that ARHL risk could be due to the additive genetic burden of a large number of variants [8], each with a small effect, or (ii) that it could be due to a limited number of very rare variants with large effect. Given the heterogeneity of ARHL, both these scenarios may be relevant, and both would escape detection by GWAS, even with very large cohort sizes. To overcome the limitations of GWAS analyses (discussed in recent reviews [6,7]), whole exome sequencing has recently been applied to ARHL population studies [5,9,10]. These studies have uncovered novel candidate genes, among which there are a number of rare, large-effect variants. Very interestingly, Mendelian hearing loss genes were among the rare variants, indicating that different mutations of the same gene may either be monogenically involved in syndromic deafness or constitute a risk factor for adult-onset hearing loss [5,10]. These results are consistent with a highly polygenic genetic architecture displaying a shared etiology between Mendelian and complex inheritance, with rare and common variants contributing to the ARHL risk.

Given the high prevalence of ARHL (estimated at 50% of people over 75 years of age-[11]) and its consequent social cost, a complete understanding of the genetic basis of this heterogeneous pathology is an important goal that could lead to the development of pharmacological and/or cell-based therapies. Attainment of this goal will require further large-scale population analyses and animal studies, as well as investigations of single individuals or families. Our current study contributes to the latter group of investigations: we report the case of two siblings affected by progressive high frequency hearing loss, which they noted in their middle-age, and which culminated in profound hearing loss in their seventies. Genetic analysis revealed that both siblings are heterozygotes for variants of two genes that are linked to Usher syndrome. This is a rare autosomal recessive disorder associated with sensorineural hearing impairment at birth, later progressive visual loss due to retinitis pigmentosa, and sometimes vestibular dysfunction [12,13]. Three genetic subtypes of this syndrome, linked to sixteen loci and thirteen identified genes, are currently recognized [14].

Our investigation revealed in both the proband and her sister a very rare truncating variant in *USH1G* and a previously undescribed missense substitution in *ADGRV1*, two genes linked to type I (OMIM 606943) and type II (OMIM 605472) Usher syndrome, respectively. The protein products of the two genes belong to the same functional network, which is involved in stereocilium biogenesis, hair bundle organization, and mechanoelectrical transduction in the auditory receptor cells of the inner ear [15], and we suggest that the two defective alleles may act additively to produce the debilitating deafness of the two siblings.

## 2. Patient and Materials and Methods

### 2.1. Subject

A 73-year-old female with progressive bilateral high frequency hearing loss and minimal benefit from hearing aids presented to the Audiology Unit of the Policlinico Ca’ Granda (Milan). The patient reported having noticed a mild hearing deficit around the age of 60. The audiogram (Figure 1) was consistent with a sensory phenotype [16], and comparison with previous audiograms carried out in different centers showed a progressive worsening of the deficit from the age of sixty-five years onward. The air-bone gap was absent. No leisure or working noise exposure was reported. No vestibular symptoms or dizziness was reported, and an ocular and vestibular evaluation gave normal responses. Familial febrile seizures were reportedly absent. The word recognition score was 70% at 80 dB HL and 10% at 90 dB HL for the left and right ear, respectively. A brain MRI revealed no abnormalities. Auditory evoked potential analysis showed a synchronous ABR peak V at low frequency, with a threshold of 60 and 50 dB HL for the right and left ears, respectively; promontory electrical stimulation tests of the auditory nerve demonstrated satisfactory residual function of both auditory nerves. The patient was referred to genetic counseling, and subsequently received an implant in the right ear.

During the following years, the hearing loss in the contralateral ear decreased further (Figure 1), and the patient recently underwent surgery for a second implant.

### 2.2. Genetic Analyses

After genetic counseling and informed consent, genomic DNA was extracted from patients’ saliva samples using the Maxwell^®^ RSC Stabilized Saliva DNA kit (Promega, Milan, Italy), a diagnostic clinical exome Next Generation Sequencing (TruSight One Expanded 6794 genes panel and DNA Prep with Enrichment by Illumina (San Diego, CA, USA)) was performed on the proband (Figure 2B, subject II,2) using the NextSeq500 Illumina platform, and primary analysis was performed with the Dragen software (version 4.2, Illumina). After filtering of common variants with general population frequency >1%, an unbiased prioritization was performed using HPO phenotypic descriptors; in addition, a selection of 78 genes involved in non-syndromic hereditary deafness (*ACTG1, ADGRV1*, *AIFM1*, *ATP6V1B1*, *BSND*, *CCDC50*, *CDH23*, *CEACAM16*, *COL4A6, COL11A2, CIB2*, *CLDN14*, *CLRN1*, *COCH*, *CRYM*, *DCDC2*, *DFNA5*, *DFNB31*, *DFNB59*, *DIABLO*, *DIAPH3*, *ESPN*, *ESRRB*, *EYA4*, *GIPC3*, *GJB2*, *GJB3*, *GJB6*, *GRHL2*, *GRXCR1*, *HARS1*, *HGF*, *HOMER2*, *ILDR1*, *KARS*, *KCNQ4*, *KITLG*, *LHFPL5*, *LOXHD1*, *LRTOMT*, *MARVELD2*, *MET*, *MIR96*, *MSRB3*, *MYH9*, *MYH14*, *MYO1A*, *MYO15A*, *MYO3A*, *MYO6*, *MYO7A*, *NARS2*, *OTOA*, *OTOF*, *PCDH15*, *PDZD7*, *POU3F4*, *POU4F3*, *PRPS1*, *PTPRQ*, *RDX*, *SERPINB6*, *SLC17A8*, *SLC26A4*, *SLC26A5*, *SMPX*, *TBC1D24*, *TECTA*, *TNC*, *TJP2*, *TMC1*, *TMIE*, *TMPRSS3*, *TPRN*, *TRIOBP*, *USH1C*, *USH1G*, *USH2A*) was analyzed using the bioinformatic tools enGenome-eVai and Alamut-Sophia Genetics, as well as the databases: NCBI dbSNP (https://www.ncbi.nlm.nih.gov/snp/ (accessed on 20 May 2023)), gnomAD (https://gnomad.broadinstitute.org/ (accessed 20 May 2023 on)), ClinVar ([17]-https://www.ncbi.nlm.nih.gov/clinvar (accessed on 20 May 2023)), LOVD (https://www.lovd.nl/ (accessed on 20 May 2023)), PubMed (https://pubmed.ncbi.nlm.nih.gov/ (accessed on 20 May 2023)), and Mastermind-Sophia Genetics (https://www.sophiagenetics.com (accessed on 20 May 2023)). Variants were classified according to ACMG-AMP criteria [18].

The presence of the variants identified by NGS was confirmed by direct Sanger sequencing. After amplification, PCR products of the genetic variant and surrounding regions were purified using Clean PCR (CleanNA-PH, Waddinxveen, The Netherlands) and sequenced in both directions using a Big Dye Terminator v.1.1 Cycle Sequencing Kit (Applied Biosystems, Foster City, CA, USA). Sequencing products were purified using a Big Dye X-Terminator Kit (Applied Biosystems, Foster City, CA, USA) and run on an ABI 3730 Genetic Analyzer (Applied Biosystems, Foster City, CA, USA). Called sequences were aligned to the reference using Sequencer V.5.0 Software (Gene Codes Corporation, Ann Arbor, MI, USA).

### 2.3. Structural Predictions

To analyze the effect of the missense mutation I4473N in the ADGRV1 structure, amino acid sequences corresponding to the 3rd, 13th, 22nd, 30th, and 32nd repeats of the Calx-β domain were submitted to either ColabFold [19] for structural predictions using the AlphaFold2-MMseqs2 algorithm [20] or to Rosetta for structural predictions using the RoseTTAFold algorithm [21]. Structures were examined using PyMOL (The PyMOL Molecular Graphics System, Version 2.0. Schrödinger LLC). Effects of the mutation on protein stability were predicted with DynaMut2 [22], Polyphen [23], SIFT [24], and CADD [25].

## 3. Results

### 3.1. Genetic Analysis

A genetic basis for the severe ARHL of the proband was suggested by the discovery that her sister, three years her senior, was affected by severe progressive sensorineural hearing loss as well. The proband’s DNA was therefore subjected to NGS sequencing on an Illumina platform (see Section 2), which revealed the presence in heterozygosis of both a rare variant (frequency 0.0008%) of *USH1G* (also known as *SANS*, NM_173477.5: c.1162G>T), and a previously undescribed variant of *ADGRV1* (also known as *VLGR1* or *GPR98*, NM_032119.4: c.13418T>A). The nucleotide substitution in *USH1G* generates a truncating stop codon (p.E388X), whereas the one in *ADGVR1* is a missense variant (p.I4473N). No other variants associated with hereditary deafness were detected among the 78 analyzed genes. The novel *ADGRV1* variant was absent from a population of 2400 subjects with pathologies not involving hearing loss analyzed in our laboratory.

To confirm the presence of the variants identified by NGS and to investigate whether the same mutations were carried by the proband’s sister, we analyzed the DNA of the two siblings by using direct Sanger sequencing (Figure 2). This analysis confirmed the NGS results and—remarkably—revealed the presence of the same two mutations in heterozygosis in the proband’s sister.

**Figure 2 biomedicines-11-02657-f002:**
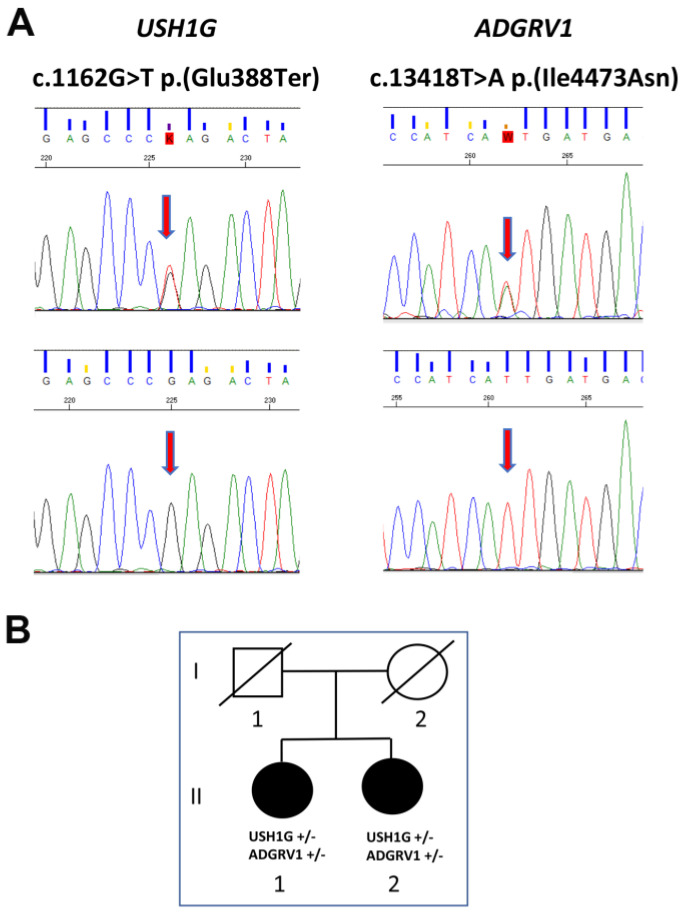
Pedigree and direct Sanger sequencing. (**A**) Upper panel: electropherograms of subjects’ *USH1G* and *ADGRV1* genes show the same heterozygous nucleotide substitutions. The lower panels show electropherograms of wild type sequences. The position of the two nucleotide substitutions are indicated by the vertical red arrows. (**B**) Lower panel: Relationship between the two subjects. In the pedigree, filled symbols indicate the presence of the clinical phenotype; empty symbols indicate a healthy status.

Both *USH1G* and *ADGRV1* are linked to Usher Syndrome (annotated as # 606943 and # 605472, respectively, in the OMIM database), a rare autosomal recessive disorder characterized by profound congenital hearing impairment and subsequent retinitis pigmentosa [12,13]. In the following parts of this report, we describe the structure and function of the two gene products, their interrelationship, and the predicted consequences of the observed nucleotide substitutions in more detail.

### 3.2. USH1G

The *USH1G* gene codes for a 481-residue protein are also known as SANS (Scaffold protein containing ankyrin repeats and SAM domain) because of the presence of three ankyrin repeats in the N-terminal portion and a SAM (Sterile alpha motif) towards the C-terminus (Figure 3A). These domains, in addition to a PDZ-binding motif at the C-terminus, confer to USG1G/SANS the capacity to undergo multiple protein–protein interactions, and USH1G/SANS is indeed a cytosolic scaffolding protein that functions within the Usher protein network [26]. USH2G/SANS-containing complexes are localized within hair cell stereocilia bundles, where they mediate the indirect interaction between transmembrane adhesive proteins (Cadherin 23 and Protocadherin 15) and the actin filaments that constitute the core of the stereocilium (reviewed in [15]). The complexes are thought to play a pivotal role at the stereocilium tip lateral links, which are key players in mechanoelectrical transduction [27,28] Mice lacking a functional USH1G/SANS gene product are deaf and present a disrupted cohesion of stereocilia with a reduction of mechanoelectrical currents [27,29]

Since the discovery [30] of *USH1G* as causally linked to the most severe form of Usher syndrome (type I-[14]), several pathogenic variants of this gene in affected families have been identified, among which are eight nonsense mutations like the c.1162G>T variant reported here (Figure 3A). This variant is present with a very low frequency in large population datasets (8 × 10^−6^), where it has so far been identified in heterozygotic subjects (gnomad.broadinstitute.org (accessed on 15 May 2023)), and not in Usher Syndrome subjects. The resulting gene product is predicted to lack the C-terminal SAM domain and PDZ binding motif. Loss of the SAM domain due to frameshift mutations is sufficient to cause loss of function and pathogenicity in mice [29] and humans [30]. Based on these considerations, the c.1162G>T transversion can be classified as a class 5 mutation (very strong evidence of pathogenicity according to ACMG criteria [18]).

### 3.3. ADGRV1

ADGRV1 (Adhesion G-Protein Coupled Receptor V1)—also known as: GPR98 (G-protein coupled receptor 98; VLGR1 (Very large G-Protein coupled receptor 1); Ush2C (Usher syndrome type-2C protein); and MASS1 (Monogenic audiogenic seizure susceptibility protein 1 homolog)—is a member of the Adhesion G-Protein coupled receptor family, composed of membrane proteins that couple intercellular or cell-matrix adhesion to intracellular signaling pathways [31,32]. With its 6306 residues, ADGVR1’s b isoform is the largest known cell surface protein [33]. The majority of its mass is contained in the enormous N-terminal extracellular domain, which comprises 35 repeats of the Calx-β domain (Pfam PF03160), one laminin G/pentraxin domain, and seven epitempin/epilepsy-associated repeats (EPTP/EAR) [34]. The extracellular domain is followed by the canonical seven membrane-spanning helices of G-protein-coupled receptors and an intracellular tail, which contains features for interaction with heterotrimeric G proteins as well as a C-terminal PDZ-binding motif (Figure 3B).

The Calx-β domain is characterized by a typical immunoglobulin fold featuring two Ca^2+^-binding sites [35]. This domain is present in cytosolic portions of some membrane proteins, as in Na^+^/Ca^2+^ exchangers, in which it serves as a Ca^2+^ sensor [35]; It is also present in the extracellular region of adhesion proteins as in the case of ADGRV1, where it is thought to mediate homotypic, Ca^2+^-dependent, protein–protein interactions [34]. In hair cells of the inner ear, ADGRV1, like USH1G, is involved in intercilium links underlying hair bundle organization. In mice deleted for the ADGRV1b isoform, hair bundles become progressively disorganized after birth, and both inner and outer hair cells are lost, with concomitant hearing impairment [34,36,37].

Since the discovery of *ADGRV1* as a causal gene of type II Usher syndrome [38], a large number of variants have been identified, and, with varying strength of evidence, they have been linked to syndromic or non-syndromic hearing loss as well as to epilepsy (e.g., see https://www.hgmd.cf.ac.uk/ac/index.php (accessed on 30 March 2023); https://www.ncbi.nlm.nih.gov/clinvar (accessed on 15 May 2023)). Among the reported variants are a number of single nucleotide missense substitutions, many of which fall within a Calx-β domain. As a matter of fact, and as illustrated in Figure 3B, nearly every Calx-β domain is a target of at least one missense mutation reported to be pathogenic, highlighting the importance of each of these 35 repeats for ADGRV1’s function. The p.I4473N substitution reported here falls into the 30th Calx-β domain (Figure 3B) and is not present in any genetic database.

To gain insight into the possible pathogenicity of the I4473N substitution, we first analyzed the degree of conservation of the variant residue. We therefore carried out multiple sequence alignments (MSA) of the thirty five human ADGRV1 Calx-β domains (Figure 4), of the 30th Calx-β domain of a number of vertebrate species (Alamut Visual Plus -Appendix A), and also examined the Calx-β seed alignment (Pfam PF03160—Appendix A). In all cases, the Ile residue aligning with I4473 is moderately conserved, and the corresponding position is predominantly occupied by a hydrophobic residue. Notably, the region to which the mutation maps is among the most conserved regions of ADGRV1’s Calx-β domains, consisting of hydrophobic residues followed by an acidic region (Figure 4).

To investigate how the substitution of a hydrophobic residue with a polar one might alter the folding of the 30th Calx-β domain, we modeled the structure of the wild-type and mutant domain with Colabfold [19] and RoseTTAFold [21] structural prediction algorithms (Figure 5). Both tools predict position 4473 to lie immediately downstream to β strand E at the beginning of the loop that leads to the C-terminal β-strand F (Figure 5A,B). Both tools predict some changes in interactions involving the residue at position 4473 with upstream residues in the loop connecting to strand D: compared to I4473, N4473 can form an additional polar bond with an upstream G4447 or V4448 for ColabFold or RoseTTAFold predictions, respectively (Figure 5C,D), but its presence disrupts a hydrophobic cluster normally formed by I4473 and I4451 (Figure 5E). In addition, as illustrated in Figure 5F, the change of I4473 to N is predicted to create a steric clash with G4447.

To be noted, the two Asp residues downstream to I4473 (D4474 and D4475) as well as D4449, which form a polar bond with I or N4473, align with residues shown to contribute to the two Ca^2+^ binding sites in the Calx-β domains of the mammalian Na^+^/Ca^2+^ exchanger [35]. Thus, the I4473N substitution could impair Ca^2+^ binding. In addition, stability prediction by Dynamut2 [22] suggests that the mutation decreases the stability of the domain by 0.22 kcal/mol when modeled using the structure obtained with ColabFold. We asked whether a similar destabilization is predicted for likely pathogenic variants in other ADGRV1 Calx-β domains, in which the variant residue lies close to the position that aligns with position 4473 in the 30th domain. We screened the Alamut Visual Plus database (Sophia Genetics) for ADGRV1 variants classified as likely pathogenic by ACMG criteria and falling within a window of 31 residues centered on the aligned 4473 position. For comparison, we also analyzed a variant in an analogous position classified as likely benign. Appendix A illustrates the positions of the selected positions relative to I4473 in the 30th domain. As shown in Table 1, all the likely pathogenic variants, but not the likely benign one, were predicted to have decreased stability. Likewise, Polyphen, SIFT, and CADD all predicted detrimental effects of the substitutions in the likely pathogenic variants, but differently from DynaMut2, they failed to distinguish between the likely beneficial variant and those classified as likely pathogenic (our unpublished observation).

Among the analyzed variants, the p.A3155D mutation is of particular interest because residue 3155 in Calx-β domain 22 is in almost the same position as residue 4473 in domain 30. Both A3155 and I4473 are predicted to have hydrophobic interactions with amino acids from the previous adjacent loop, and this hydrophobic cluster would be disrupted by a substitution with a polar or charged residue.

In summary, the absence of the I4473N substitution from population databases (PM2), its co-segregation with deafness in the two siblings (PP1), its presence in a well-established functional domain (PM1), the moderate conservation of the residue, its large Grantham Distance from the substituting Asn, as well as the structural predictions (PP3) classify the new variant described here as likely pathogenic (class 4, Table 1).

## 4. Discussion

We report here the simultaneous presence in heterozygosis of a pathogenic and a likely pathogenic variant in two Usher-syndrome-linked genes in two siblings affected by progressive severe to profound sensorial hearing loss after the age of seventy. Although Usher Syndrome is a recessively inherited disease, we consider it likely that the combined presence of the two variants underlies the profound ARHL reported here. Indeed, the *USH1G* and the *ADGRV1* gene products belong to the same functional network involved in stereocilium organization and function in cochlear hair cells, and in retinal photoreceptors [15,47]. As illustrated in Figure 6, string analysis with *USH1G* as the query places the two proteins in the same protein network (Figure 6A), and Gene Ontology enrichment analysis shows them to be involved in common pathways (Figure 6B). The USH protein network consists of multiprotein complexes that integrate cell adhesion molecules (among which ADGRV1) with intracellular scaffolding proteins (among which SANS) and, via unconventional Myosin 7A, with the actin cytoskeleton. In this way, intercilium adhesion is coupled to the rigid actin filament core of stereocilia, and the striking architecture of cochlear auditory receptor hair bundles is generated (reviewed in [15,48]).

Although ADGRV1’s function has been best characterized in intercilium ankle links [49] and SANS is known to be essential for the tip links [27,28], all Usher proteins can essentially directly or indirectly interact with each other via their PDZ domains and PDZ interacting motifs [50]. Ankle links are transient structures that, in mice, are no longer present at P12 [49]; however, ADGRV1 has been detected in the stereocilia of the adult cochlea and ADGRV1 and SANS co-localize to the ribbon synapses of hair cells too [51]. Importantly, biochemical analysis of an Usher protein complex located between the inner and outer segments of mammalian photoreceptors directly demonstrated a physical interaction between SANS and ADGRV1 mediated by the scaffolding protein whirlin (USH2D) [47]. Thus, the participation of the two genes in shared pathways (Figure 6B) could result in mutual dependency of each one on the other’s dosage, and the phenotypic effects reported here could be explained by the phenomenon of combined or complex haploinsufficiency [52]. So far, we have not identified additional families with isolated ARHL associated with this kind of digenic inheritance, nor have patients with this combination of gene defects been reported by others. Nevertheless, in support of our hypothesis is the well-documented case of digenic inheritance of Type II Usher syndrome in a subject carrying heterozygous frameshift mutations in the *ADGRV1* and the *PDZD7* (OMIM 612971) modifier genes [53]; additionally, cases of likely digenic inheritance involving combinations of monoallelic variants of USH1 and USH2-linked genes have been reported as well [43].

Usher syndrome is classically a combined disorder of the retina and the cochlea; however, neither of the two siblings of the present report presented visual problems, other than those normally associated with age (cataract). This is not surprising, as Usher Syndrome retinal pathology is variable and of later onset than hearing loss [54]. Furthermore, different mutations of USH genes may be linked to non-syndromic deafness and not to full-blown Usher syndrome [12,55]. Indeed, of the SANS and ADGRV1 mutations listed in the Human Gene Mutation Database (https://digitalinsights.qiagen.com/products-overview/clinical-insights-portfolio/human-gene-mutation-database/ (accessed on 9 March 2023)), more are reported as being linked to isolated deafness than to Usher Syndrome.

The current study was limited to two siblings affected by near complete hearing loss, as other family members were not available for genetic testing. The father and mother (deceased at the time of this study) died at the ages of seventy and of eighty three, respectively, and neither of them ever complained of hearing problems. To the knowledge of the siblings, there were no other reports of severe hearing loss among known relatives. Thus, a likely scenario is that each of the two variant alleles was carried in heterozygosis by a single parent, and that the siblings received one variant from each parent.

In conclusion, our study reveals a novel genetic basis for the progressive severe to profound sensory deafness of two siblings, confirming the complexity of genotype–phenotype correlations of *USH* genes, and providing a previously undescribed example of digenic inheritance of ARHL. Hopefully, the case reported here will provide a useful addition to the growing number of investigations into the risk factors and genetics of this debilitating condition.

## Figures and Tables

**Figure 1 biomedicines-11-02657-f001:**
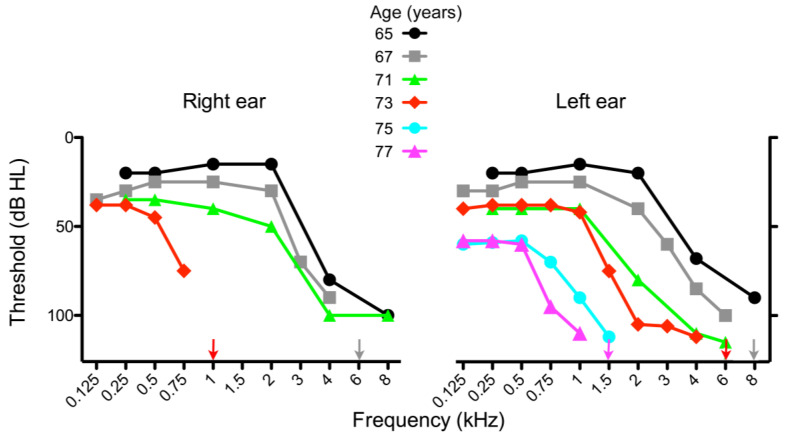
Audiograms of the subject at different ages, as indicated. Loss of hearing at high frequencies progressed more rapidly in the right ear than in the left ear. The patient received cochlear implants in the right and left ears at the age of 74 and 79, respectively.

**Figure 3 biomedicines-11-02657-f003:**
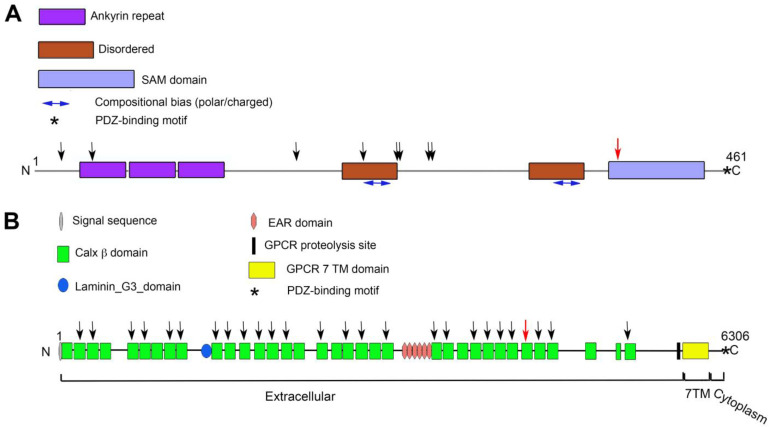
Schematic representation of domain organization of SANS (**A**) and ADGRV1 (**B**) proteins, with positions of the mutations identified in this study. In (**A**), the black vertical arrows indicate the positions of truncating mutations reported in the HGMD database. The red vertical arrow indicates the position of the p.E388X mutation identified in the subject and her sister. In (**B**), the black vertical arrows indicate the Calx β domains (green rectangles) reported in the HGMD database to carry at least one missense mutation associated with a pathology. The red vertical arrow indicates the 30th Calx β domain that carries the p.I4473N mutation reported here. Note the difference in scale between the two panels.

**Figure 4 biomedicines-11-02657-f004:**
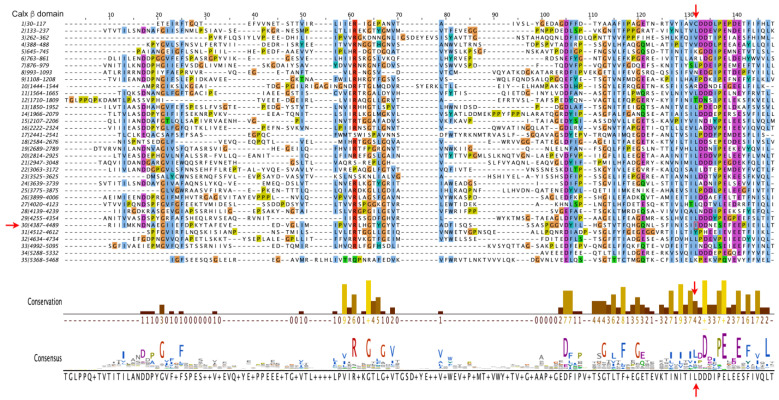
Multiple Sequence Alignment of the 35 Calx β domains of human ADGRV1. Alignment was performed with the default parameters of Clustal Omega (https://www.ebi.ac.uk/Tools/msa/clustalo/ (accessed on 23 January 2023)). Residue coloring is according to the Clustal X Color Scheme (https://www.jalview.org/help/html/colourSchemes/clustal.html (accessed on 23 January 2023)). The conservation plot [39] and the consensus sequence below the alignment were generated using the Jalview analysis package. The shading of the bars from brown to yellow reflects the conservation number indicated below the bars. The vertical red arrows indicate the position of the I4473, which the conservation plot shows to be moderately conserved and predominantly occupied by a hydrophobic residue. The 30th Calx β domain is indicated by the red horizontal arrow to the left of the alignment and the Ile substituted by Asn in the subject is boxed.

**Figure 5 biomedicines-11-02657-f005:**
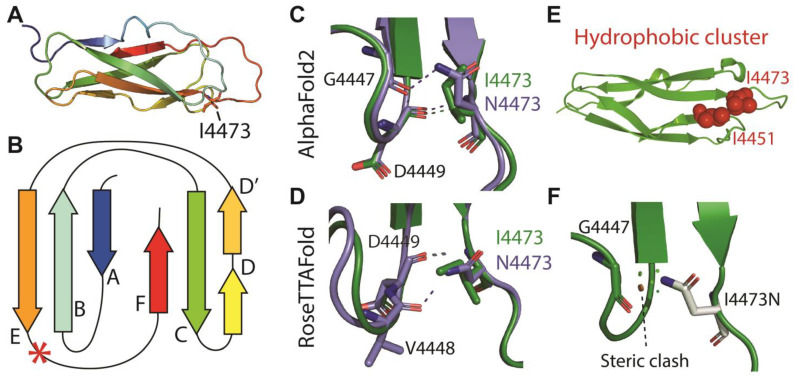
Predicted 3D structures of the wild-type 30th Calx-β domain and of the I4473N variant. (**A**): Ribbon representation of the ColabFold predicted structure of the wild-type 30th Calx-β domain colored from blue (N-terminus) to red (C-terminus). (**B**): Two-dimensional cartoon of the Calx-β domain indicating the position of the I4473 residue (red asterisk). (**C**,**D**): Comparison of the polar bonds formed by Ile or Asn 4473 in the ColabFold (**C**) or RoseTTAFold (**D**) predicted structures. (**E**): Hydrophobic cluster formed by I4473 and I4451 in the ColabFold prediction (Representation obtained with the ProteinTools toolkit [40]). (**F**): Steric clash predicted to occur between N4473 and G4447 (Representation obtained with PyMOL). The indicated red disk represents the larger clash, and smaller green disks represent smaller clashes. Note that clashes are not predicted for the wild-type domain.

**Figure 6 biomedicines-11-02657-f006:**
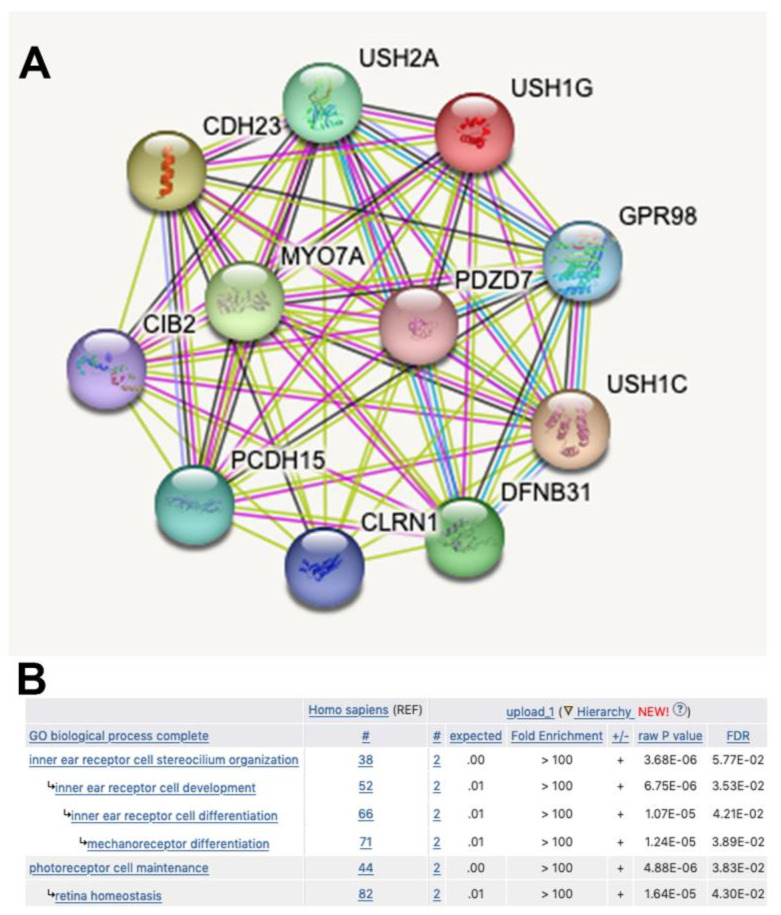
Common functional network for SANS and ADGRV1 proteins. (**A**): The String database (https://string-db.org/ (accessed on 15 May 2023)) was interrogated with USH1G. ADGRV1 is annotated as GPR98 and included together with USH1G in the USH protein network. (**B**): The GO enrichment analysis tool (http://geneontology.org/docs/go-enrichment-analysis/ (accessed on 15 May 2023)) was interrogated with the USH1G and ADGVR1 couple (UniProt identification numbers Q495M9 and Q8WXG9).

**Table 1 biomedicines-11-02657-t001:** Missense substitutions in Calxβ domains located within 15 residues from the aligned position of the I4473 mutation (see Appendix A) *.

Protein HGVS	cDNA HGVS	Calxβ Domain N.	Grantham’s Distance	GnomAD Maximum MAF	ΔΔG^Stability^ (kcal/mol)-DynaMut2 Prediction ^§^	ACMG Class	ACMG Evidence	Reported Clinical Phenotype
p.P352T	c.1054C>A	3	38	0.005% NFE	−0.25	4	PM1; PM5; PP1; PP3; PP5	USH2C [41]
p.P352L	c.1055C>T	3	98	0.0046% NFE	−0.26	4	PM1; PM5; PP1; PP3; PP5	Non-syndromic hearing loss [42]
*p.D1944N*	*c.5830G>A*	*13*	*23*	*0.5% NFE 1.5%Ashkenazi*	*0.16*	*2*	*PM1*; *PP3*; *BS1; BP6*	*USH3/hearing loss* [43,44]
p.A3155D	c.9464C>A	22	126	Not present	−1.59	4	PM1; PM2; PM3; PP3	USH2 [45]
**p.I4473N**	**c.13418T>A**	**30**	**149**	**Not present**	**−0.22**	**4**	**PM1; PM2; PP3; PP1**	**Non-syndromic hearing loss (present study)**
p.D4707Y	c.14119G>T	32	160	Not present	−0.55	4	PM1; PM2; PP3; PP5	Usher [43]
p.P4720L	c.14159C>T	32	98	Not present	−0.55	4	PM1; PM2; PP3; PP5	Usher [46]

* The variant reported in the present study is in bold; a variant classified as likely benign is in italics. ^§^ Based on the ColabFold predicted structures.

## Data Availability

The de-identified sequencing data for the two participants of this study can be accessed from the co-corresponding author, P.C. The clinical data will be available at the Audiology Unit, Ospedale Maggiore Policlinico, Milan, after obtaining authorization from the subject.

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
