# Peer review of "Combined Presence in Heterozygosis of Two Variant Usher Syndrome Genes in Two Siblings Affected by Isolated Profound Age-Related Hearing Loss"

_biomedicines, 2023, doi:10.3390/biomedicines11102657_

Round 1
Reviewer 1 Report
The authors have studied what seems to be late-onset hearing loss case in two siblings showing heterozygous variants in two known autosomal recessive hearing loss genes. These genes are usually associated with early onset/prelingual AR HI. The subjects are well-described and analyzed. The authors have supported their findings well and drawn adequate conclusions to the study. I have the following comments:
1. Please correct any minor syntax and formatting errors. Ex: font sized in figure captions in Suppl material, reference Praveen et al, 2022 listed as such and not “10”, Refer to Fig 1 when first talking about audiogram (Line 93), several word spacing issues, “Sequencher” (Line135) etc
2. This is a case report since only one family is reported. It will be interesting to see if there are more such possibly digenic families in the same region as this family hails from.
3. Was the patient 65yr old when first audiogram was conducted as stated in Fig 1? Are there audiograms from the older sibling as well for these ages to include? I would suggest to report that in the text of the Methods also to.
4. Since both the genes are known and widely studied, I would suggest the authors to significantly cut down on description of the protein structures of both known genes to keep the focus more on the effect of the variants on both the genes.
5. Have the authors considered adding the variants in public repository such as ClinVar as Pathogenic? If yes, then an accession number can be included in the manuscript.
Author Response
Reviewer N. 1:
We thank the reviewer for his/her positive comments. Below, we describe how we have addressed the specific criticisms.
- The referee asks us to correct minor syntax and formatting errors, including the fonts of the figure captions in the supplementary material, an error in a citation, the position in the text for reference to a figure and word spacing issues.
Response: We have carefully checked the ms for syntax and formatting errors. Some of the spacing issues are caused by long strings of letters that provide links to URLS (databases and analysis tools) that were used in our study. We expect that this formatting problem will be addressed by the in-house editor. The fonts of Fig. S1 have been adjusted to match those of Fig. S2, the reference Praveen et al has been fixed, and Fig. 1 is now referred to when the audiogram is first mentioned.
- The reviewer notes that it will be interesting to see whether other digenic families are present in the same region of origin of the two siblings of this case report.
Response: This is a relevant comment, and we would indeed be very interested in discovering whether the variant combination that we have identified is present in other families. In fact, however, we have not so far found additional digenic families among those analyzed in our center. We might mention that the origins of the two siblings are quite complex: a Sicilian father, and a German mother with Jewish and Portuguese-Brazilian (!) ancestors. We added a sentence on the possibility of other families with the identified variant combination in the revised ms (p. 11, first paragraph): “We have not so far identified additional families with isolated ARHL associated with this kind of digenic inheritance, nor have patients with this combination of gene defects been reported by others. Nevertheless, in support of our hypothesis...."
- The reviewer asks (i) whether the patient had audiograms previous to the age of 65; (ii)whether audiograms from the older sibling are available to be included in the ms.
Response: Concerning point (i), the patient did have audiograms before the age of 65, as she began to notice some mild hearing loss around the age of 60, however, these audiograms were done in another center, she no longer has copies of them and we do not have access to them. We have now added in the text the sentence: " The patient reported that she noticed a mild hearing deficit around the age of 60." (p. 2, first paragraph of Methods section). Regarding point (ii), the older sibling was a resident of Mexico, and we have not been able to access her audiograms; as she appeared to have exactly the same hearing problem as the proband, we collected a DNA sample from her during one of her visits to Italy. She underwent cochlear implantation in Mexico City in 2019. Thus, we cannot provide any more information on her hearing loss.
- The reviewer feels that too much published information on these widely known genes is given in our manuscript and that we should instead keep the focus on the effect of the variants on both the genes.
Response: This criticism is also raised in point 4 of the second reviewer. Nevertheless, we strongly feel that the link of the biology of these two very interesting proteins to the possible effects of the two mutations is what makes the paper worthwhile and more interesting than a simple case report. Furthermore, the analysis of the effect of the p.I4473N substitution in the ADGVR1 variant can only be fully appreciated if the structure of the protein is understood. On the other hand, we realize that some material in the "Results" section is redundant with the Discussion. Therefore, we have significantly abbreviated the "Results" section, with special care to eliminate information presented in the "Discussion".
- The referee asks whether we have considered adding the variants as pathogenic in a public repository such as ClinVar.
Response: At the moment, the diagnostic data of the Laboratory are not shared with the ClinVar database. We are considering this possibility for all the results obtained through sequencing (about 2000 NGS/year, of which half are clinical exomes), not only for class 4 and 5 pathogenic variants but also for class 3 variants with unknown clinical significance. The latency is mainly due to technical and informatic limitations, which we are trying to overcome, to speed up and standardize the process of data loading.
Reviewer 2 Report
In the manuscript “Combined presence in the Heterozygosis of two Variant Usher Syndrome Genes in two Siblings affected by isolated Profound Age-Related Hearing Loss” by Borgese et al., the authors have identified a compound heterozygous mutation of USH1G (reported variant) and ADGRV1 (novel variant) genes in two siblings with age relate hearing loss. Protein structural analysis suggested the novel mutation as likely pathogenic affecting the protein structure negatively. Current finding is important in terms of unique inheritance pattern, but before publication, the reviewer suggests following modifications in the manuscript:
1. In Fig. 3A, authors have showed the position of truncating mutations with arrows. The reviewer suggests adding the mutations of ADVRG1 gene along with the arrows as well.
2. For the novel mutation of ADRVG1 gene, did authors check the presence/absence of mutation in control samples of their population?
3. The “Material & Methods” of the manuscript carry the information (including figures, clinical evaluation & sequencing results) which should be the part of “Results”. Please describe the findings of current study in results sections only.
4. In the Results section, authors have discussed the findings of previously published studies. My recommendation to the authors is to limit their results with the findings of current study only.
5. String Analysis i.e. Fig. 6, has neither been described in Material & Methods nor in Results section of the manuscript. Please add appropriate details in the relevant section.
6. Please provide the links/citations for the tools i.e. SIFT, POLYPHEN, CADD, DynaMut2, NCBI, dbSNP, ClinVar & LOVD in methodology section of the manuscript.
7. Some minor comments:
(i) In line 41, please provide citation for “Praveen 2022”
(ii) Please provide proper citation for lines 60-63
(iii) Line 320, please replace “missense” with “Missense”
(iv) Line 314, please replace “Table I” with “Table 1”
(v) Lines 370-371, please provide the link of human mutation database here
(vi) In Table 1, please explain the asterisk “*” given in the caption (see Fig. S2)*
I suggest authors to check the title of the manuscript for language correction. Proof-reading of the manuscript is also required as it needs language improvement throughout.
Author Response
The reviewer has several useful suggestions, which we have attempted to address, as detailed below.
- The reviewer suggests modifying Fig. 3, so that mutations are indicated by arrows in both panels (and not by arrows in panel A and circles in panel B).
Response: This has been done.
- The reviewer asks whether we have checked for the presence/absence of the novel ADGRV1 variant in control samples of our population.
Response: We checked for the presence/absence of the novel ADGRV1 mutation in 2400 samples analyzed with a clinical exome (TruSight ONE Expanded Illumina) with a clinical condition different from ARHL or other deafness forms: no one had the c.13418T>A variant. We have added the sentence "The novel ADGRV1 variant was absent from a population of 2400 subjects with pathologies not involving hearing loss analyzed in our laboratory." (p. 4, first paragraph of section "Genetic analyses".
- The referee notes that the Methods section contains information that should be moved to the Results section, in particular part of the clinical evaluation of the patient and sequencing results.
Response: We agree that the description of the two new mutations should not be reported in the Methods section, and have deleted this part from the Methods in the revised ms (lines 125-128 of the original ms). We feel instead that the clinical evaluation of the patient does not belong in the Results section, rather, the description of the patient represents the starting point for out study, but is not in itself a Result. Therefore, we prefer to leave the entire paragraph on the patient in the section that should be entitled "Patient and Methods" or "Patient and Materials and Methods".
- The referee notes that in the "Results" section, we discuss results of the literature, and asks us to limit the content of this section to our own novel results.
Response: This criticism is also raised in point 4 of the first reviewer. Nevertheless, we strongly feel that the link of the biology of these two very interesting proteins to the possible effects of the two mutations is what makes the paper worthwhile and more interesting than a simple case report. Furthermore, the analysis of the effect of the p.I4473N substitution in the ADGVR1 variant can only be fully appreciated if the structure of the protein is understood. On the other hand, we realize that some material in the "Results" section is redundant with the Discussion. Therefore, we have significantly abbreviated the "Results" section, with special care to eliminate information presented in the "Discussion".
- The reviewer asks us to describe String analysis in the relevant section.
Response: We have now added to the legend of Fig. 6 the link to the URL of this tool for the study of protein-protein interactions.
- The reviewer asks us to provide links/citations in the Methods section for the databases and analysis tools used in our study.
Response: This has been done; the links/citations can be seen at the end of p. 3 and on p. 4 (second paragraph) of the revised ms.
- Minor comments:
(i) provide reference for Praveen et al - Response: done
(ii) provide proper reference for lines 61-63 (concerning proportion of population affected by ARHL) - Response: done
(iii) Title of Table 1 - first word should be capitalized - Response: done
(iv) Refer to the Table with arabic numeral (Table 1) - Response: done
(v) Provide the link for the Human Gene Mutation Database in the Discussion - Response: done (p. 11, second paragraph of revised ms).
(vi) Explain the meaning of the asterisk in the caption of Table 1 - Response: the asterisk refers to the first footnote at the bottom the table, the footnote explains the meaning of the bold and italics characters for two of the entries.
Comments on the quality of the English Language:
Response: the article has been proofread by an English mother-tongue speaking colleague, so we hope that the language is OK.